# High-Resolution Optical Remote Sensing Image Registration via Reweighted Random Walk Based Hyper-Graph Matching

**Yingdan Wu** [1,2,3] **, Liping Di** [3,*] **, Yang Ming** [4] **, Hui Lv** [1,2] **and Han Tan** [5]

[1]  School of Science, Hubei University of Technology, No. 28 Nanli Road, Wuhan 430068, China;
    yd_wu2010@hbut.edu.cn (Y.W.); lvhui@hbut.edu.cn (H.L.)
[2]  Hubei Collaborative Innovation Centre for High-efficient Utilization of Solar Energy,
    Hubei University of Technology, No. 28 Nanli Road, Wuhan 430068, China
[3]  Center for Spatial Information Science and Systems, George Mason University, Fairfax, VA 22030, USA
[4]  Institute of Surveying and Mapping, CCCC Second Highway Consultants Co., Ltd., No. 18 Chuangye Road,
    Wuhan 430056, China; mingyang@ccshcc.cn
[5]  Wuhan Vocational College of Software and Engineering, No. 117 Guanggu Avenue, Wuhan 430205, China;
    fanzhihai_2003@mail.hzau.edu.cn
**\***  Correspondence: ldi@gmu.edu; Tel.: +1-703-993-6114

**Abstract:** High-resolution optical remote sensing image registration is still a challenging task due to non-linearity in the intensity differences and geometric distortion. In this paper, an efficient method utilizing a hyper-graph matching algorithm is proposed, which can simultaneously use the high-order structure information and radiometric information, to obtain thousands of feature point pairs for accurate image registration. The method mainly consists of the following steps: firstly, initial matching by Uniform Robust Scale-Invariant Feature Transform (UR-SIFT) is carried out in the highest pyramid image level to derive the approximate geometric relationship between the images; secondly, two-stage point matching is performed to find the matches, that is, a rotation and scale invariant area-based matching method is used to derive matching candidates for each feature point and an efficient hyper-graph matching algorithm is applied to find the best match for each feature point; thirdly, a local quadratic polynomial constraint framework is used to eliminate match outliers; finally, the above process is iterated until finishing the matching in the original image. Then, the obtained correspondences are used to perform the image registration. The effectiveness of the proposed method is tested with six pairs of high-resolution optical images, covering different landscape types—such as mountain area, urban, suburb, and flat land—and registration accuracy of sub-pixel level is obtained. The experiments show that the proposed method outperforms the conventional matching algorithms such as SURF, AKAZE, ORB, BRISK, and FAST in terms of total number of correct matches and matching precision.

**Keywords:** high-resolution optical remote sensing imagery; image registration; reweighted random walk; hyper-graph matching

## 1. Introduction

Image registration is to align two or more images taken by different sensors or by the same sensor but from different viewpoints [1,2], which is the basis processing of many remote sensing applications, such as change detection [3], image fusion [4], and environment monitoring [5]. For high resolution optical remote images, the relationship between intensity value of conjugate pixels is complex, which is known as non-linear differences and large geometric distortion [6], making the

radiometric characteristics of local images differ from each other. Features appearing in one image may not be present in another one, and vice versa. Therefore, registration of high-resolution optical remote imagery is still a challenging task.

The process of image registration can generally be divided into three phrases: image matching, transformation model establishing, and image resampling. Because of geometric deformation induced by image rotation, different scales and viewing angles, as well as aforementioned non-linear differences of intensity, finding corresponding features is difficult [7], especially for the multi-senor remote sensing imagery. There are two categories for the methods of image matching: area-based matching (ABM) and feature-based matching (FBM) [8]. Normalized correlation coefficient (NCC) and mutual information (MI) are the common similarity measures in the ABM methods, but they are sensitive to image intensity changes and geometric deformations. When directly applying them to the multi-sensor satellite imagery matching, it is hard to obtain the desirable results. A great deal of FBM algorithms have been developed in the past three decades, and various feature detection algorithms have been proposed, such as Harris [9], BRISK [10], ORB [11], and MSER [12]. For the feature description, SIFT [13], PCA-SIFT [14], BFSIFT [15], GLOH [16], LSS [17], and so on have been developed. The Euclidean distance of description vectors is often used to evaluate the similarity between the features.

For the establishment of transformation model, the affine and polynomial transformation models are commonly used. However, these methods can only roughly estimate the transformation for the remote sensing image usually having large size and complex geometric distortion. To overcome this problem, local methods such as triangulated irregular network (TIN) [18], B-spline [19], and thin-plate splines [20] are developed and they often outperform the global methods. However, due to the complex nature of multi-sensor image matching, a large number of false matches would exist and should be removed. RANdom Sample Consensus (RANSAC) [21] is the mostly used methods to remove outliers in the matching points, and researchers have developed several RANSAC-like algorithms such as maximum likelihood estimation sample consensus [22], and maximum distance sample consensus [23]. Kouyama et al. and Sugimoto et al. utilized observation geometry as a constraint in their RANSAC steps and achieved robust performance of outlier rejection even under the cloudy condition [24,25], but the discrimination ability of this kind of algorithms largely depends on the geometric transformation model. The interpolation methods of bilinear interpolation or cubic convolution interpolation are frequently used to warp the input image to the reference image by using the estimated parameters of the image transformation model [26].

Among feature-based matching methods, SIFT and its modified versions are successfully applied for the multi-sensor satellite image matching in recent years. Schwind et al. [27] skipped the first octave to improve the time efficiency and enhance the matching precision at the same time by reducing the number of noise feature from the first octave. Suri et al. [28] assigned a uniform orientation for all features to suppress the orientation computation. Saleem and Sablating [29] used the normalized gradient SIFT for matching multispectral images, Hasan et al. [30] used the neighborhood information, and Yi et al. [31] and Mehmet et al. [32] modified the gradient of SIFT and set restriction on scale changes to refine the matching precision. However, all these feature descriptors are hand-crafted by the designer's expertise or intuition, and it is hard to cover all the different situations, especially for remote sensing imagery. Recently, deep learning techniques are more and more utilized to describe the feature points detected from the remote sensing images [33,34]. However, all these feature-based matching algorithms only depending on the intensity similarity of feature points, or the spatial similarity of only nearby features, they are prone to produce incorrect matches.

However, based on the observation that despite large intensity changes of images captured from different viewpoints or by different sensors, the structure among them remains relatively stable. Therefore, structural information could be taken advantage of to improve matching robustness. Liu et al. [35] introduced a multi-stage matching approach, in which firstly homograph transformation between the reference image and the searching image is estimated by initial matching, and then probability relaxation is used to expend matching. Many researchers cast the image matching problem

as the graph matching problem, such as spectral graph matching [36], probabilistic graph matching (HGM) [37,38], balanced graph matching [39], tensor-based high-order graph matching (TM) [40], and reweighted random walks graph matching [41,42]. However, in these algorithms, the number of graph nodes is only a few dozen, the computational cost is very high and a large amount of computer memory is required, which could be unaffordable for the remote sensing image registration.

On the basis of previous studies, this paper presents a reweighted random walk based hyper-graph matching for registration of multi-sensor optical remote-sensing imagery, and it mainly has two main characteristics:

(1) Improving the robustness and success rate of image matching without paying too high of a computational cost. Image matching is an ill-posed problem, and ABM, FBM, as well as graph matching methods have their pros and cons. Currently, most graph matching methods have high computation cost and require large amount of computer memory, and many of them are not suitable for the remote sensing image registration since it needs to match a large number of feature points. This paper describes a framework of image matching that integrates ABM, FBM and graph matching methods together to improve the image matching robustness and success rate without paying too much computation cost;

(2) Simultaneously utilizing high-order structure information and one-order intensity similarity in the matching process in an efficient way. Taking building the three-order similarity tensor for example, most graph matching algorithms will randomly sample a certain number of triangles for each point in the reference image, and all the possible triangles will be selected. In this paper, the candidates for each matching feature point are firstly searched by the ABM method, and the feature points' candidate relationship is utilized to build the hyper-edge tensor, which can significantly improve the sparseness of association graph and the computational efficiency.

## 2. Methodology

### 2.1. Matching Feature and Process

#### 2.1.1. Matching Feature

Various features—such as point features, linear features, and regional features—could be utilized. In the case of multi-sensor remote sensing imagery registration, although the linear and regional features are more stable and easy to identify and match, the extracted features often appear to be fragmented, incomplete, or not completely coincidental. Moreover, homogenous linear features and areal regions do not guarantee existing or having a well distribution in the image. All these factors are not conducive to the image registration of high accuracy. For point features, they are not limited to the image content and more suitable for the general cases. Through developing suitable matching strategies to enhance the matching success rate and robustness, it could guarantee the correct matching of point features, and when hundreds and thousands of feature points are successfully matched, they can depict the local differences accurately and derive satisfactory results.

Therefore, the point feature is selected as the matching primitive. Strategies—such as initial point position prediction for false matching points, rotation and scale invariant area-based matching, and hyper-graph matching—are integrated to obtain large amount of well-distributed feature points.

#### 2.1.2. Matching Process

The framework of our image registration method is illustrated in Figure 1. Before starting matching, the image pyramid with three levels is generated from original image in which different levels have different spatial resolutions (see Section 2.2). Firstly, well-distributed feature points were extracted by the Förstner operator. Next, the initial matching was conducted by SIFT matching method, and the affine transformation between the matching images was computed. Then, a rotation and scale invariant area-based matching method is used to derive the candidate matching points.

Finally, the hyper-graph matching which can simultaneously consider the radiometric and high-order structure information is used to find the correct matches from the matching candidates. Local quadratic polynomial constraint framework and TIN-based image resampling are adopted to eliminate the outliers and perform accurate image registration. The coarse-to-fine strategy is adopted and the matching results from the higher pyramid level are used as guidance for the matching in the next pyramid level.

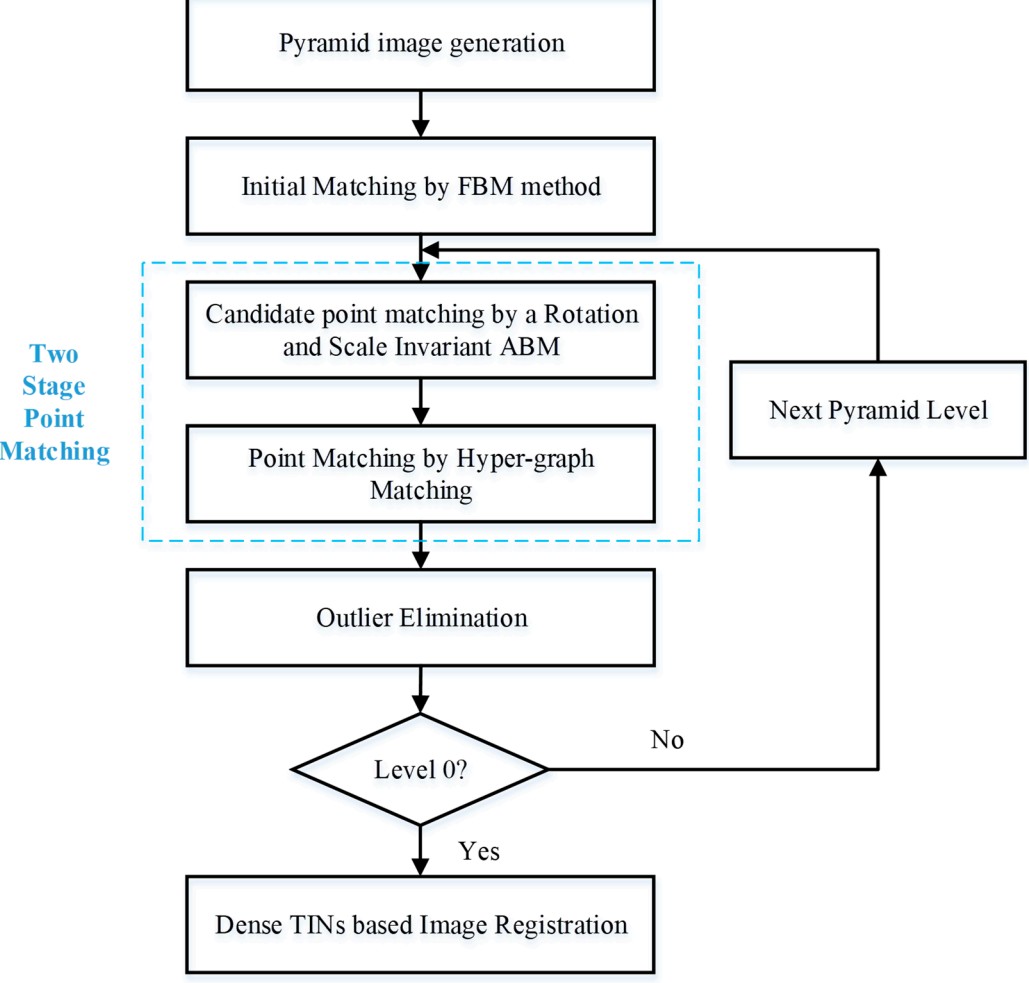

**Figure 1.** Workflow of our registration method.

## 2.2. Initial Matching by FBM Method

Since the size of remote sensing imagery is usually large, the coarse-to-fine strategy is adopted to increase the matching efficiency. The pyramid image is generated by the $3 \times 3$ pixel average method. The original image is regarded as the pyramid image of level zero, then the average gray value of $3 \times 3$ pixels in the original image is assigned to the corresponding pixel of the pyramid image of level one, iterating this process until a pyramid image of level three is generated.

The initial matching is only carried out in the pyramid image of highest level. The main purpose of initial matching is deriving the approximate geometric relationship between the reference image and the searching image. With this relationship, we can obtain the approximate values about the rotation and scale differences between the images and the range of overlapping area. In this paper, UR-SIFT [43] is employed for the initial matching. Compared with SIFT, UR-SIFT feature would more likely retain high-quality features across the entire image. The ratio R between the Euclidean distance to the closest neighbor and that of the second closest is set to be 0.7.

In order to robustly estimate the parameters of transformation model, the RANSAC algorithm is used to eliminate the effect of false matches. Since the geometric deformations are very small for the pyramid image of the highest level, the affine transformation model is sufficient to describe the geometric constraint between the matches

$$\begin{cases} x_2 = h_{11} + h_{12}x_1 + h_{13}y_1 \\ y_2 = h_{21} + h_{22}x_1 + h_{23}y_1 \end{cases} \tag{1}$$

where $(x_1, y_1)$ are the coordinates of a point in the reference image, $(x_2, y_2)$ are the coordinates of its corresponding point in the searching image, $h_{ij}$ is the parameters of affine transformation model, and $h_{11}, h_{21}$ represent the offset, $h_{12}, h_{13}, h_{22}$ and $h_{23}$ reflect the rotation and scale between the images. After obtaining the affine transform coefficients, the rotation angle $\theta$ and the scale $\lambda$ between the images can be calculated using following formula:

$$\begin{cases} \theta = \frac{1}{2} \times (atan(\frac{h_{13}}{h_{12}}) + atan(-\frac{h_{21}}{h_{22}})) \\ \lambda = \frac{1}{2} \times ( \sqrt{h_{12}^2 + h_{13}^2} + \sqrt{h_{22}^2 + h_{23}^2}) \end{cases} \tag{2}$$

After the initial matching, the scope of overlapping area can be determined, which is divided into virtual grid cells. For each grid cell, one feature point is extracted in it by the Förstner operator.

### 2.3. Two-Stage Point Matching

In order to improve the success rate and robustness, a two-stage point matching approach is employed. Firstly, an ABM method—a rotation and scale invariant area-based matching algorithm is used to search the candidate matching point; and secondly, the high-order structural information between the feature points are exploited to find the correct matching points.

### 2.3.1. Candidate Point Matching by a Rotation and Scale Invariant ABM

For the multi-sensor optical satellite image registration, there would be large rotation and scale changes between the images, and this would lead to the failure of traditional NCC method. Thus, the correlation windows are warped before matching in the presence of such geometric distortions.

Specifically, we open a matching window in the reference image, predict its corresponding center position in the searching image, perform the image matching window warping, and finally calculate the similarity. The specific steps are as follows:

(1) Prediction of initial position of conjugate point in the searching image. In the pyramid image of highest level, affine transformation model is directly used to calculate the initial position of conjugate point in the searching image for each point in the reference image. In other levels, for the points successfully matched in the higher level, just directly project it to the current level. For the point failed to be matched in the higher level, its nearest successfully matched point is used to derive the initial position.

As shown in Figure 2, $(P_1, P_1')$ is a point pair successfully matched in the higher pyramid level, and $P_1$ is the nearest successfully matched feature point for $P_2$, which fails to match in the higher pyramid level. Based on the knowledge that structural information is relatively stable, through $(dx, dy)$, which is the image distance between the feature point $P_1$ and $P_2$, the rotation angle $\theta$ and the scale $\lambda$ calculated from formula (2), we can obtain the initial position of the conjugate point of $P_2$ by the following formula

$$\begin{cases} x_2' = x_1' + \lambda \times (dx \cos \theta - dy \sin \theta) \\ y_2' = y_1' + \lambda \times (dx \sin \theta + dy \cos \theta) \end{cases} \tag{3}$$

where $(x_1', y_1')$ is the image coordinates of point $P_1'$, $(x_2', y_2')$ is the predicted image coordinates of point $P_2'$.

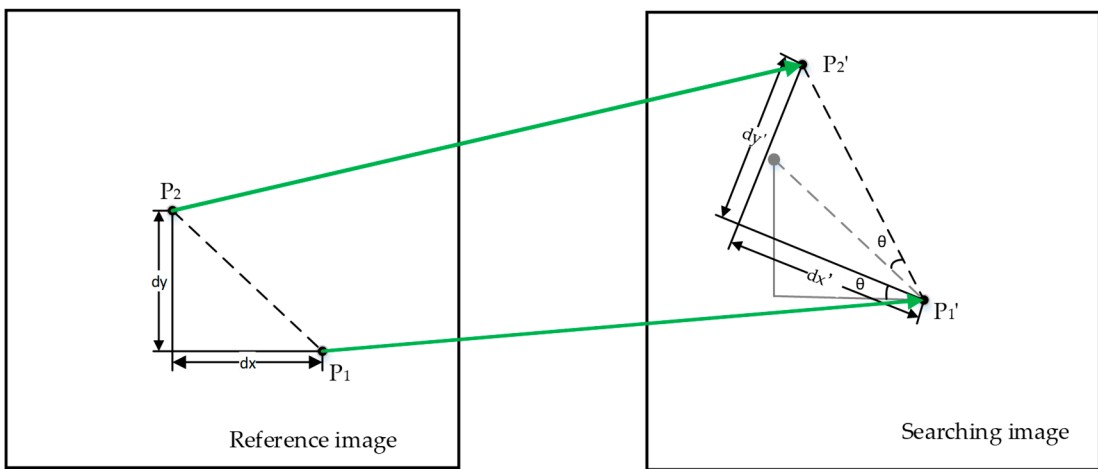

**Figure 2.** Initial position prediction for points failed to match in the higher pyramid image level: $(P_1, P'_1)$ is a point pair successfully matched in the higher pyramid level, $P_2$ is a feature point failed to matching in the higher level, $P'_2$ is the initial position predicted by our method. $\theta$ is the rotation angel between the reference image and the searching image. $(dx, dy)$ is the distance between the feature point $P_1$ and $P_2$ in the image. $(dx', dy')$ is the calculated distance between the $P'_1$ and with consideration of the rotation and scale difference between two matching images. Since the $P_1$ is successfully matched and the position of $P'$ is known, we can derive the initial position for the feature point $P_2$ although it was failed to match previously.

(2) Perform the image matching window wrapping. The image matching window in the searching image is rotated and scaled according to the rotation angle $\theta$ and scale parameter $\lambda$. By doing the image matching window wrapping, the geometrical distortion effects can be partially compensated.

### 2.3.2. Point Matching by Hyper-Graph Matching

After finding candidate points in the searching image, the next step is to find correspondences between two sets of features, this process can be defined as a graph matching problem. This kind of approach is usually restricted to the normal graph embedding unary and pairwise relations. Pairwise relations are not enough to incorporate the information about the entire geometrical structure of features. Embedding higher-order information into the matching will overcome the limitation of pairwise similarity. Recent hyper-graph matching methods incorporate higher-order similarity measures to achieve more accurate results.

A hyper-graph $G = (V, E, A)$ consists of nodes $v \in V$, hyper-edges $e \in E$, and attributes $\alpha \in A$ associated with the hyper-edges. We consider two feature sets $P$ and $Q$, which can formulate two hyper-graphs $G^P = (V^P, E^P, A^P)$ and $G^Q = (V^Q, E^Q, A^Q)$, The goal of the hyper-graph matching is to establish mapping between nodes of two hyper-graphs $G^P = (V^P, E^P, A^P)$ and $G^Q = (V^Q, E^Q, A^Q)$. $n_p$ and $n_Q$ denote the numbers of node in $G^P$ and $G^Q$, respectively. We do not assume $n_p = n_Q$, i.e., there may be different numbers of feature points in the two feature sets to be matched. In the case of tie point matching, $n_Q$ is usually larger than $n_p$. Suppose a set of all possible node correspondence $C = V^P \times V^Q$, and $k$-tuples $c_{w1} = (v^P_{p_1}, v^Q_{q_1}), \ldots, c_{wk} = (v^P_{p_k}, v^Q_{q_k}) \in C$ among them. For $k$-th order hyper-graph matching, the similarities of the $k$-tuples are measured by comparing attributes of two $k$-th order hyper-edges $e^P_{p_1, \ldots, p_k}$ and $e^Q_{q_1, \ldots, q_k}$, which mean the hyper-edge connecting $v^P_{p_1}, \ldots, v^P_{p_k}$ and $v^Q_{q_1}, \ldots, v^Q_{q_k}$ respectively. The similarity function is denoted by $\Omega(\cdot, \cdot)$, the $k$-th order similarity of the

$k$-tuple is measured by $\Omega_k(\alpha^P_{p_1,\ldots,p_k}, \alpha^Q_{q_1,\ldots,q_k})$, so the $k$-th order similarity tensor $H^{(k)}$ can be derived in a recursive manner as

$$
\begin{aligned}
H^{(k)}_{c_{w_1},\ldots,c_{w_k}} &= \Omega_k(\alpha^P_{p_1,\ldots,p_k}, \alpha^Q_{q_1,\ldots,q_k}) \\
&+ \lambda^{(k-1)} \sum_{l=1}^{k} H^{(k-1)}_{\{c_{w_1},\ldots,c_{w_k}\}\setminus c_{wl}}, \\
H^{(1)}_{c_{w_i}} &= \Omega_1(\alpha^P_{p_i}, \alpha^Q_{q_i})
\end{aligned}
\tag{4}
$$

Assuming the value of maximum order among all hyper-edges is $\delta$, the resulting similarity tensor $\mathbf{H}^{(\delta)}$ contains the entire similarity information. In the rest, we abbreviate $\mathbf{H}^{(\delta)}$ as $\mathbf{H}$.

The problem of hyper-graph matching is equivalent to looking for a binary assignment matrix $X \in \{0,1\}^{n_P \times n_Q}$ such that $\mathbf{X}_{p,q}$ is equal to 1 when $v^P_p \in V^P$ matches to $v^Q_q \in V^Q$, and to 0 otherwise. It is natural to force one-to-one constraints that make $\mathbf{X}$ a permutation matrix

$$
\mathbf{X}\mathbf{1}_{n_Q \times 1} \leq \mathbf{1}_{n_p \times 1}, \mathbf{X}^{\mathsf{T}}\mathbf{1}_{n_p \times 1} \leq \mathbf{1}_{n_Q \times 1}
\tag{5}
$$

where $\mathbf{1}_{n \times 1}$ denotes an all-ones vector with size $n$. Given $\mathbf{x}$ is the vector version of matrix $\mathbf{X}$, the hyper-graph matching score is defined as

$$
S(\mathbf{x}) = \sum_{c_{w_1},\ldots,c_{w_k}} H_{c_{w_1},\ldots,c_{w_k}} \mathbf{x}_{c_{w_1}} \cdots \mathbf{x}_{c_{wk}}
\tag{6}
$$

Here the product $\mathbf{x}_{c_{w_1}} \cdots \mathbf{x}_{c_{wk}}$ will be equal to 1 if the points $\{p_1, \ldots, p_k\}$ are all matched to the points $\{q_1, \ldots, q_k\}$, and 0 otherwise. In the first case, it will add $H_{c_{w_1},\ldots,c_{w_k}}$ to the total score function. This is a similarity measure, which will be high if the sets of features $\{p_1, \ldots, p_k\}$ is similar to the set $\{q_1, \ldots, q_k\}$. We can find that $\mathbf{H}$ contains similarity values of all the order, $S(\mathbf{x})$ amounts to the summation of all similarity values in all order, then the goal of the hyper-graph matching is to find the assignment vector $\mathbf{x}^*$ which maximizes the matching score function $S(\mathbf{x})$ under the constraints of Equation (5)

$$
\mathbf{x}^* = \underset{\mathbf{x}}{\mathrm{argmax}} S(\mathbf{x})
\tag{7}
$$

Several algorithms have been proposed to solve this problem, including tensor matching (TM) [40], reweighted random walks for hyper-graph matching method (RRWHM) [41], hyper-graph matching (HGM) [42], etc. In this letter, a modified RRWHM method is introduced with consideration of the candidate relationship among the feature points.

First, an association graph $G^w = (V^w, E^w, A^w)$ is defined, in which every node represents a candidate correspondence. A random walk from a node $v_{w1} \in G^w$ to another node $v_{w2} \in G^w$ on the

graph

$$
\begin{aligned}
d_w &= \sum_{w_2,\ldots,w_\delta} \mathbf{H}_{w,w_2,\ldots,w_\delta} = (\mathbf{H} \otimes \mathbf{1} \ldots \otimes \mathbf{1})_w \\
d_{\max} &= \max_w d_w \\
\mathbf{P} &= \mathbf{H}/d_{\max} \\
x^{(t+1)} &= \mathbf{P} \otimes_2 x^{(t)} \ldots \otimes_\delta x^{(t)} + (1-\lambda)\mathbf{r}
\end{aligned}
$$

$G^w$ implies a walk from a correspondence $c_{w1}$ to another correspondence $c_{w2}$ between $G^P$ and $G^Q$. To address the problem of scaling up the outlier nodes weight, an absorbing node is added to $G^w$ to perform the affinity-preserving random walks, the transition tensor $\mathbf{P}$ and the updating rule for the hyper-graph random walks can be summarized as

$$
\begin{aligned}
d_w &= \sum_{w_2,\ldots,w_\delta} \mathbf{H}_{w,w_2,\ldots,w_\delta} = (\mathbf{H} \otimes \mathbf{1} \ldots \otimes \mathbf{1})_w \\
d_{\max} &= \max_w d_w \\
\mathbf{P} &= \mathbf{H}/d_{\max} \\
x^{(t+1)} &= \mathbf{P} \otimes_2 x^{(t)} \ldots \otimes_\delta x^{(t)} + (1-\lambda)\mathbf{r}
\end{aligned}
\tag{8}
$$

In the equation, $d_w$ is the degree of the node represented by the sum of the weight values associated with it, $x^{(t)}$ is $x$ at $t$-th iteration, **r** is the vector for personalization, $\lambda$ is the bias parameters, and **1** is the all one vector.

In the method of RRWHM, to build the three order similarity tensor **H**, first a certain number of triangles per points in reference image are randomly sampled, and all possible triangles in the searching image are sampled and their descriptors are calculated and stored by KD-tree. For each of selected triangles in the reference image, several hundred, such as 300, nearest neighbors in the searching image are searched to build **H**. It can be found that in the RRWHM, building tensor **H** is very time consuming and needs large amount of computer memory. For image registration, the number of feature points would be a few thousand, building three-order tensor would demand too much memory that regular computer cannot afford.

To solve this problem, a sparse high-order similarity tensor without losing any useful structure information is built. After the candidate points are derived by the rotation and scale invariant ABM matching method, when vertex of the triangle in the reference image and that of in the searching image is not the correspondence candidate, then these two triangles would definitely not become the corresponding hyper-edges. Therefore, when we set the maximum number of candidate points for each feature point as 5, then the number of candidate triangles is only $5^3$ at most, and there is no need to calculate all possible triangles in the searching image but only sample a few hundred nearest triangles to serve as the candidate triangles for building the tensor **H**.

Due to the image distortion, pairwise similarity measure, which is sensitive to the scale change, is not used in this study. Only the unary similarity and three-order similarity measure are adopted. For the unary similarity in **H**, NCC value is directly used. For the three-order similarity, three angles in the triangle are compared with their sine values

$$\mathbf{H}_{w_1, w_2, w_3} = exp\left(-\frac{1}{\sigma_s}\sum_{k=1}^{3}\left|\sin(\theta_{w_k}^P) - \sin(\theta_{w_k}^Q)\right|\right) \tag{9}$$

where $\theta_{wk}^P$ and $\theta_{wk}^Q$ are the angles of node related to the correspondence $w_k$. A balance weight is used to merge the unary similarity tensor $\mathbf{H}^{(1)}$ and triple similarity tensor $\mathbf{H}^{(3)}$,

$$\mathbf{w}_H = \frac{\sum_{i=1}^{n_1} f_1}{\sum_{i=1}^{n_2} f_3} \tag{10}$$

where $f_1$ is the unary similarity, $f_3$ is the triple similarity, $\mathbf{w}_H$ is the balanced weight, $n_1$ is the number of matched points and $n_2$ is the number of triangles. In this way, geometric and radiometric information are simultaneous utilized in the matching process.

## 2.4. Outlier Elimination and Image Resampling

In the matching process, the outliers are inevitably present, and they must be eliminated before undergoing the image registration. Traditionally, RANSAC-like algorithms are used to estimate the transformation model, such as homography and quadratic polynomial. The matches not conforming with the transformation model are regarded as outliers. However, when there is a relatively complex image distortion, the transformation model cannot serve as a criterion to determine outliers. Therefore, a local quadratic polynomial constraint framework is proposed in this letter. The quadratic polynomial is shown as

$$\begin{cases} x_s = a_0 + a_1 x_m + a_2 y_m + a_3 x_m y_m + a_4 x_m^2 + a_5 y_m^2 \\ y_s = b_0 + b_1 x_m + b_2 y_m + b_3 x_m y_m + b_4 x_m^2 + b_5 y_m^2 \end{cases} \tag{11}$$

where $(a_i, b_i)(i = 0, \cdots, 5)$ are the polynomial coefficients, $(x_m, y_m)$ and $(x_s, y_s)$ are the image coordinates of the matched point pairs in the reference image and searching image respectively.

The procedure of outlier elimination is as follows:

(1)　Adopt the kd-tree to store the image coordinates of the matching points.

(2)　Traverse and judge each matching point. For the current judging point, several nearest neighboring points around it are collected by using the K-NN strategy on the basis of image coordinates distance. For quadratic polynomial is used, we recommend the number of nearest neighboring points is better larger than 10. The estimated quadratic polynomial is used to calculate the coordinate residual of current judging point. When the coordinate residual is greater than RMSE twice, the judge point is regarded as outliers and indexed.

(3)　Return to step (1) to reconstruct kd-tree using the matching points which are not labeled as outlier after traversing all matching points.

(4)　Iteratively perform above process until no matching point is labeled as outlier.

The retained matching points are used as control points (CP) to form a pair of dense TINs to describe the mapping function. For each pair of triangles, the affine transformation is calculated, which is used to resample the image of the triangle in the searching image to the reference image. Iterating to process all the triangles, the image registration between the reference image and searching image is completed.

## 3. Experiments and Analysis

### 3.1. Description of Test Data

Six pairs of images captured by WorldView-1/3, GeoEye, GF-1, and SPOT-5 sensors are used for the evaluation. These datasets are all formed by the high-resolution optical satellite remote sensing images, covering different land type, such as mountain area, urban, suburb, and flat land. The images of the last two datasets are also covered by some clouds. The detail information of the experimental datasets is presented in Table 1.

**Table 1.** Detail information of experimental datasets.

| No. | Platform | Acquisition Time | Land Type | Image Size (Pixels) | Pixel Size (m/Pixel) |
|-----|----------|------------------|-----------|---------------------|----------------------|
| 1 | WorldView-1 | 2009 | Mountain | $35,180 \times 11,028$ | 0.5 |
| 2 | WorldView-1 | 2010 | Suburb | $35,154 \times 13,045$ | 0.5 |
| 3 | GeoEye | 2013 | Urban | $27,552 \times 25,132$ | 0.5 |
| 4 | SPOT-5 | 2008 | Flat | $12,000 \times 12,000$ | 5.0 |
| 5 | GF-1 | 2017 | Flat | $2000 \times 2000$ | 16 |
| 6 | WorldView-3 | 2016 | Urban | $4000 \times 4000$ | 0.5 |

### 3.2. Matching Results and Analysis

According to the matching strategy mentioned above, UR-SIFT matching is performed in the highest pyramid image level, to derive rough transformation information between the matching images. With this, the values of rotation angle and scale difference between the reference image and searching image are obtained, which are used for the compensation of searching image window. With the coarse-to-fine strategy, the matching results in the higher level are used as guidance for the matching in the next level. It is mainly manifested in the following two aspects: firstly, for the points successfully matched, just project its coordinates to the current level to derive its initial position; secondly, for the points that failed to match, search the nearest successfully matching point and use the relative geometric relationship between them to calculate its initial position in the current level.

Through this strategy, the matching failed points in the higher level can still be matched, and this would enhance the matching success rate effectively.

The block size used in our approach is 800 × 800 pixels, for each block several hundred or a few thousand feature points are extracted by the Förstner operator. The size of template window and search window is 13 × 13 pixels and 35 × 35 pixels respectively, and the threshold for normalized correlation coefficient is 0.7. After searching matched candidates for each point, the hyper graph matching is carried out block by block. As shown in the Figure 3, abundant feature points with effective distribution in all of the experimental images can be obtained.

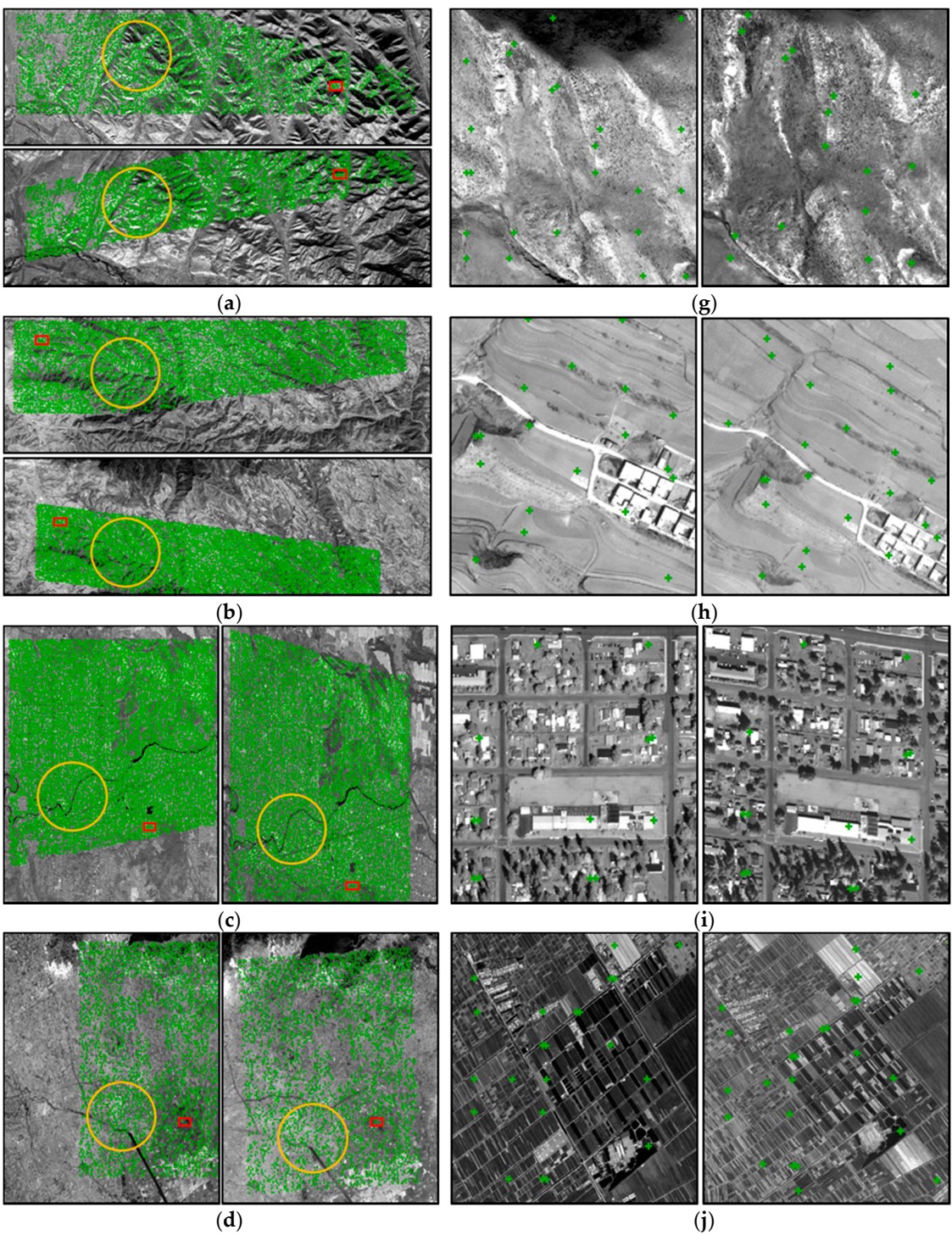

**Figure 3.** *Cont.*

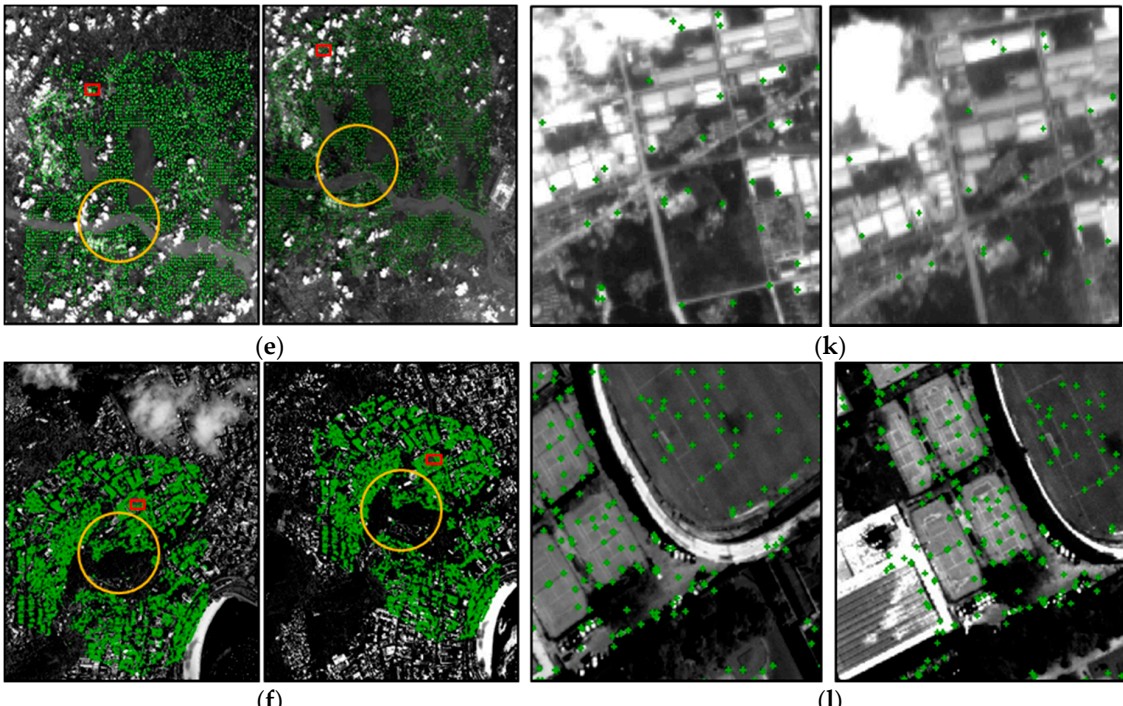

**Figure 3.** Matching results of our method: (**a**–**f**) show the matching results for the datasets 1–6 respectively, and (**g**–**l**) show the details of the sub-images marked by the rectangle in the (**a**–**f**). The red box represents the place of detail window on the right, the yellow circle represents the approximate position of the sub-images of registration chessboard images showed later in Figure 4, and the green cross represents the successfully matched corresponding points.

The first image pair is acquired from WorldView-1 satellite sensor, covering the mountainous area in Qinghai province of China with obvious illuminations differences and clear topographic relief. These factors result in locally non-linear difference in intensity and distortions in geometry. Nevertheless, our method can still obtain evenly distributed matched points.

The second image pair is also from the WorldView-1 satellite sensor, and it mainly covers suburb area containing large amount of terrace. The topographic relief is also obvious, and the repetitive texture from the terrace would make trouble to the feature point matching. However, our method can obtain satisfactory matching results as well.

The third image pair is from the Geoeye satellite sensor covering the urban areas and with a very high resolution of 0.5 m. The images are characterized with structured textures, and these textures benefit the feature point matching, but the locally geometric distortion caused by the buildings makes the accurate image registration very difficult.

The fourth image pair is from the flat area, and the image resolution is relatively low (5 m) compared with the aforementioned three datasets (0.5 m). The two images in this pair are from SPOT-5 satellite, and they contain large amount of farmland and buildings. They are captured at different seasons with large time span of over two years, and so forth the images have considerable textural changes due to the rapid development in China (as shown in Figure 3h). It can find matches successfully in the unchanged textures, and good matching results are obtained.

The fifth image pair is from GF-1 satellite, and the image resolution is relatively low (16 m) compared with the aforementioned three datasets (0.5 m). Due to the low image resolution, the geometric distortion is relatively small, but some parts of the image are covered by the clouds, which would make trouble for the accurate image registration.

The sixth image pair is from the Worldview-3 satellite sensor covering the urban areas with an image resolution of 0.5 m. For the high image resolution and tall buildings, the distortion of building is very severe, in additional, there is some clouds covering a large part of the top of the image, which makes the registration extremely hard.

To further examine the registration effectiveness of our method, the registration chessboard images of the sub-images marked by the circle in the Figure 3a–d are generated, depicted in Figure 4. The brightness of the slave image is specially adjusted to better express the registration effect.

It can be found that, for the first image pair—because the terrain is undulating and the mountain registration result is relatively poor—the main features such as mountain undulations, roads, and residential areas are accurately registered. For the image pairs of 2–4, significant linear objects such as roads and rivers have been accurately registered in suburbs, urban areas, and the continuity of image features has been well preserved. For the image pairs of 5–6, although the images covered by the clouds, which affect the distribution of the feature points, sufficient feature points can be derived in the areas not covered by the clouds, which enables us to get good registration results.

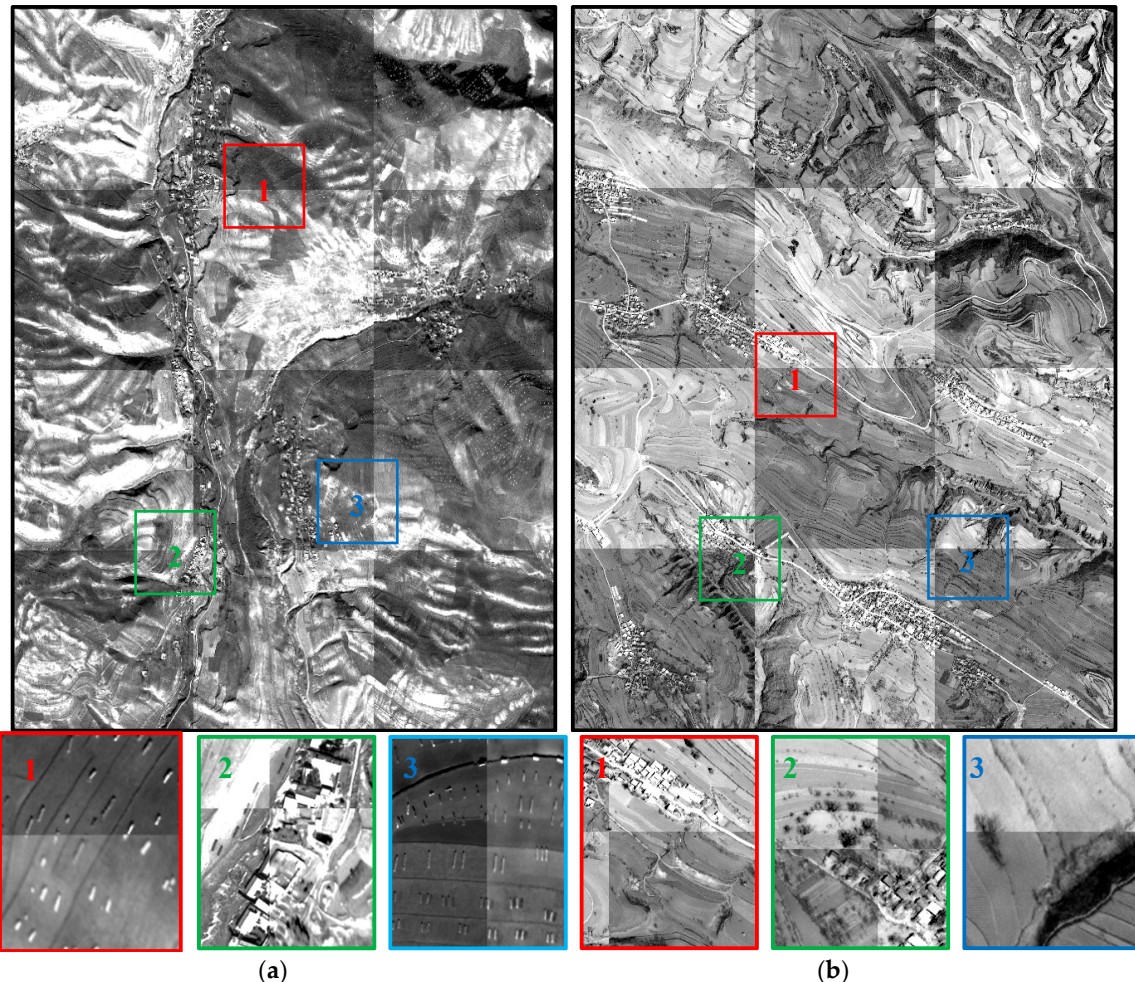

(a) (b)

**Figure 4.** *Cont.*

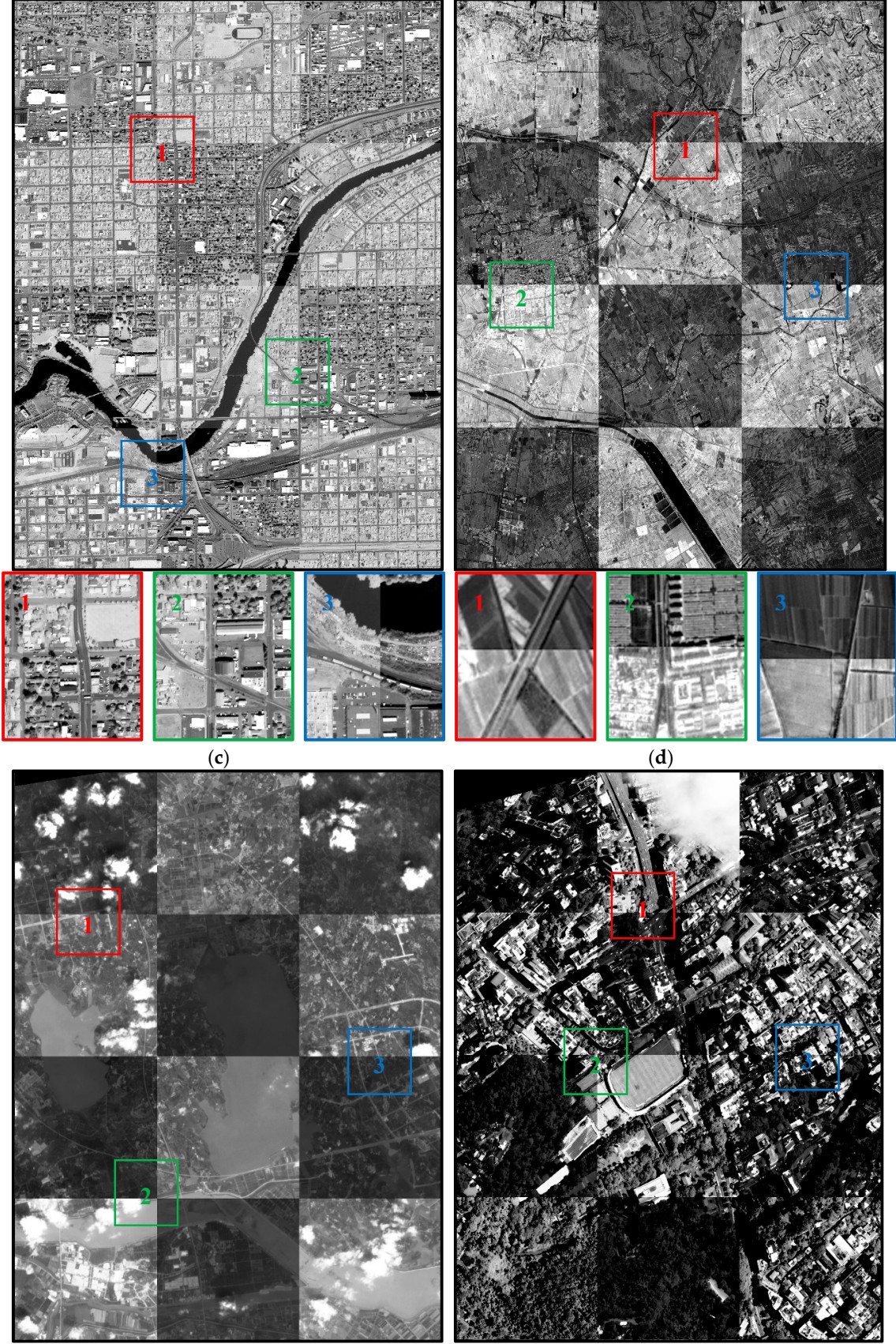

**Figure 4.** *Cont.*

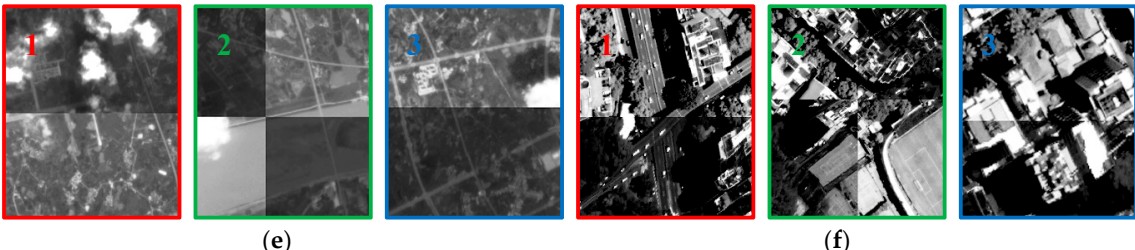

(e)            (f)

**Figure 4.** Registration results of our method: (**a**–**f**) is the registration details of the sub-images marked by the circle in the Figure 3a–f. For each dataset, three enlarged detail windows are illustrated, which are numbered with 1, 2, 3 and colored with red, green and blue accordingly.

To quantitatively analyze the registration accuracy, a number of conjugate points are selected as checkpoints in the above experimental area. The calculated coordinates of checkpoints on the slave images are obtained by interpolation using the dense TINs formed by the matching results and are compared with the correct coordinates on the slave image. The differences between the two are statistically analyzed. The registration results for the six datasets are shown in the Table 2.

**Table 2.** Registration accuracy for the six image pairs.

| Datasets | Number of Checkpoints | RMS (Pixels) | |
|:---:|:---:|:---:|:---:|
| | | x | y |
| 1 | 60 | 0.41 | 0.63 |
| 2 | 60 | 0.63 | 0.58 |
| 3 | 85 | 0.42 | 0.22 |
| 4 | 85 | 0.41 | 0.33 |
| 5 | 100 | 0.36 | 0.27 |
| 6 | 100 | 0.52 | 0.75 |

### 3.3. Comparison with Other Methods

In the paper, SURF, AKAZE, ORB, BRISK, and FAST are chosen as the competitors to evaluate the effectiveness of the method. Recall and precision [43], were used as criteria in the evaluation. M is the number of matching points, C is the number of existing correspondences, CM is the number of correctly correspondences, and the two criteria are defined as: recall = CM/C and precision = CM/M.

Since the comparing algorithms often adopt a global transformation such as projective model to eliminate the erroneous matches, which could not accurately express the geometric transformation for the image with large size, especially covering mountainous and urban areas. Thus, four pairs of sub-images with size of 800 × 800 pixels are grabbed from the four datasets are used for evaluation. For the algorithms like SURF, BRISK, and FAST, we directly used the relevant functions provide by the MATLAB 2018b with default values to perform the experiment; and for the algorithms like AKAZE and ORB, as they are not provided by the MATLAB 2018b, the open source software OpenCV with version 4.1.1 is used, and the parameters for the feature detection, description and matching are all default values. The computer configuration is as follows: Window10, Inter (R) Core (TM) i7-6700 @ 3.4.0GHz, 8G RAM. The matching results for four image pairs are shown in Table 3 and Figure 5.

As shown in the Table 3 and Figure 5, our method can obtain large number of feature points and outperform the other matching algorithms, especially in the terms of number of correct matches and matching precision. Hundreds of point pairs can be acquired by our method, while other algorithms can only obtain a few dozens of them, which are not sufficient for the accurate image registration, especially for the image with complex geometric distortion. The precision and the successfully matched points of our method are dramatically better than the other methods. However, we find that the recall

of FAST algorithm is better than our method. We find that C is the successfully matched points, and C of FAST algorithm is very small than our method; for example, C is only 63 for FAST but 1428 for our method, and larger the C number higher the false match probability, that may be the reason why FAST algorithm have a higher recall. However, the C from FAST algorithm is too small, even though it has a relatively high recall, the correctly matched points are too few to perform accurate image registration.

**Table 3.** Matching results for four pairs of sub-images.

| Image Pair | Indicators | SURF | AKAZE | ORB | BRISK | FAST | Our Method |
|---|---|---|---|---|---|---|---|
| 1 | C | 162 | 52 | 246 | 131 | 63 | 1428 |
| | CM | 53 | 21 | 84 | 59 | 51 | 1108 |
| | Recall | 32.72% | 40.38% | 34.15% | 45.04% | 80.95% | 77.59% |
| | Precision | 4.35% | 9.06% | 4.53% | 7.81% | 0.77% | 31.21% |
| | Time(s) | 0.54 | 1.57 | 3.73 | 3.30 | 1.02 | 22.64 |
| 2 | C | 295 | 81 | 223 | 60 | 119 | 2221 |
| | CM | 99 | 59 | 90 | 29 | 76 | 1246 |
| | Recall | 33.56% | 72.84% | 40.36% | 48.33% | 63.87% | 56.10% |
| | Precision | 10.07% | 15.76% | 6.45% | 8.17% | 3.20% | 46.90% |
| | Time(s) | 0.47 | 1.31 | 2.83 | 1.70 | 0.50 | 38.33 |
| 3 | C | 116 | 59 | 96 | 81 | 16 | 1234 |
| | CM | 0 | 6 | 24 | 31 | 11 | 474 |
| | Recall | 0.00% | 10.17% | 25.00% | 38.27% | 68.75% | 38.41% |
| | Precision | 0.00% | 3.20% | 1.50% | 1.02% | 0.12% | 28.34% |
| | Time(s) | 0.87 | 1.97 | 4.89 | 0.51 | 1.98 | 20.58 |
| 4 | C | 99 | 78 | 128 | 108 | 23 | 2546 |
| | CM | 8 | 38 | 31 | 37 | 17 | 1432 |
| | Recall | 8.08% | 48.72% | 24.22% | 34.26% | 73.91% | 56.25% |
| | Precision | 1.78% | 2.74% | 1.97% | 1.18% | 0.17% | 68.00% |
| | Time(s) | 0.67 | 1.55 | 4.06 | 0.98 | 1.54 | 47.64 |

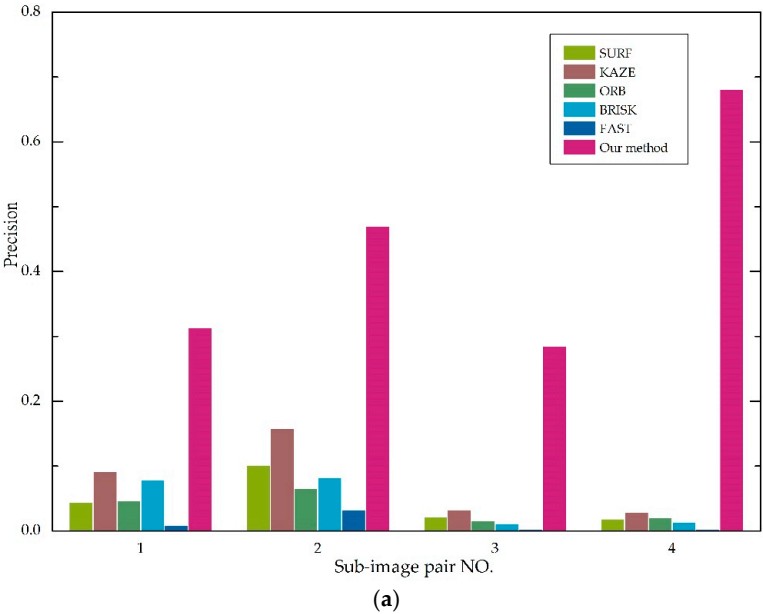

(a)

**Figure 5.** *Cont.*

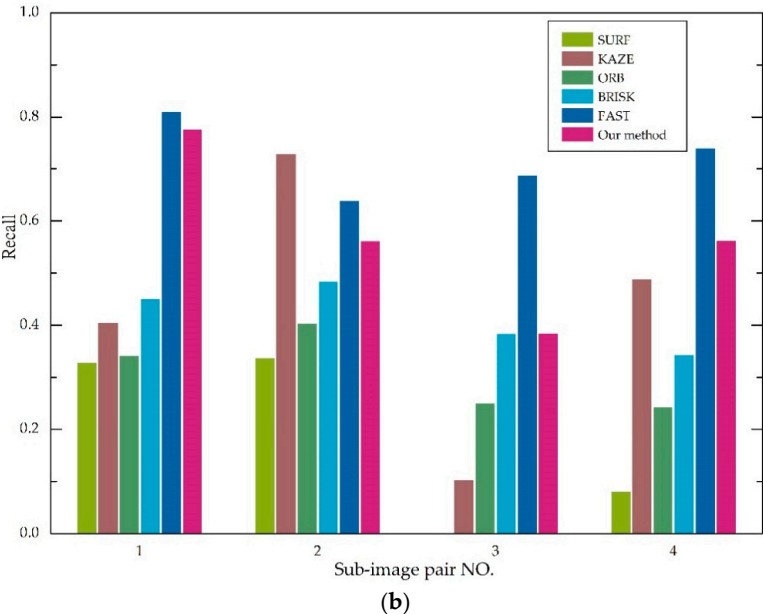

**(b)**

**Figure 5.** Quantitative matching results of four image pairs: (**a**) matching precision; (**b**) matching recall.

In general, the methods such as SURF, AKAZE, ORB, BRISK, and FAST, are carried out by nearest neighboring distance ratio of radiometric descriptors. When encountered image with repetitive textures or non-linear intensity differences, these algorithms are hard to find conjugate features, and this is the main reason that this kind of algorithms can only obtain a few feature points. For our method, it can simultaneously make use of the radiometric and high-order geometric information to search the corresponding points, which largely enhance the matching robustness and success rate, and by integrating the coarse-to-fine and outlier detection strategy, the matching in the current level will benefit from the matching results from former level, which ensures the whole matching process is robust and produces more correct points. This can be easily seen in Figure 6, which is the comparing results between our method and other classical feature matching methods for the image pair 2. Figure 7 is the matching results from our method without hyper-graph matching algorithm, which determines the correspondence for each feature point only by the rotation and scale invariant ABM. From the results, it is found that—through the strategies of rotation and scale invariant ABM, blunder rejection, and coarse-to-fine matching—it can derive large number of matches. However, for the repetitive textures of the image, it is prone to obtain false matches, and hard to reject them completely, as shown by the circle in the Figure 7. By integrating the high-order structural information in the matching process, it contributes a lot to the matching robustness and success rate.

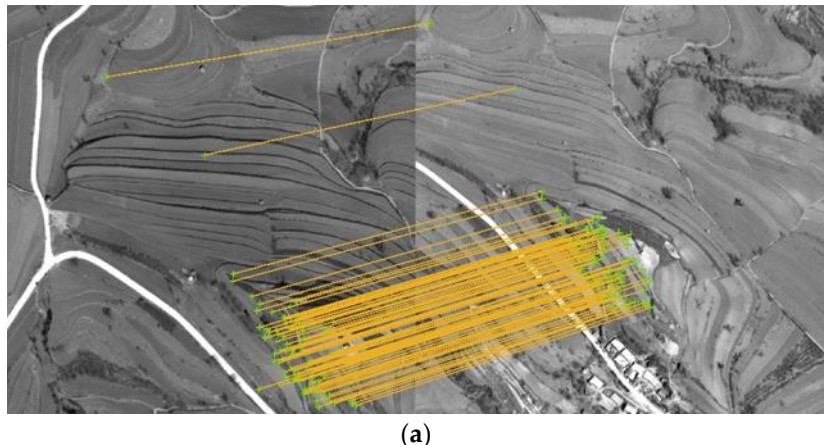

**(a)**

**Figure 6.** *Cont.*

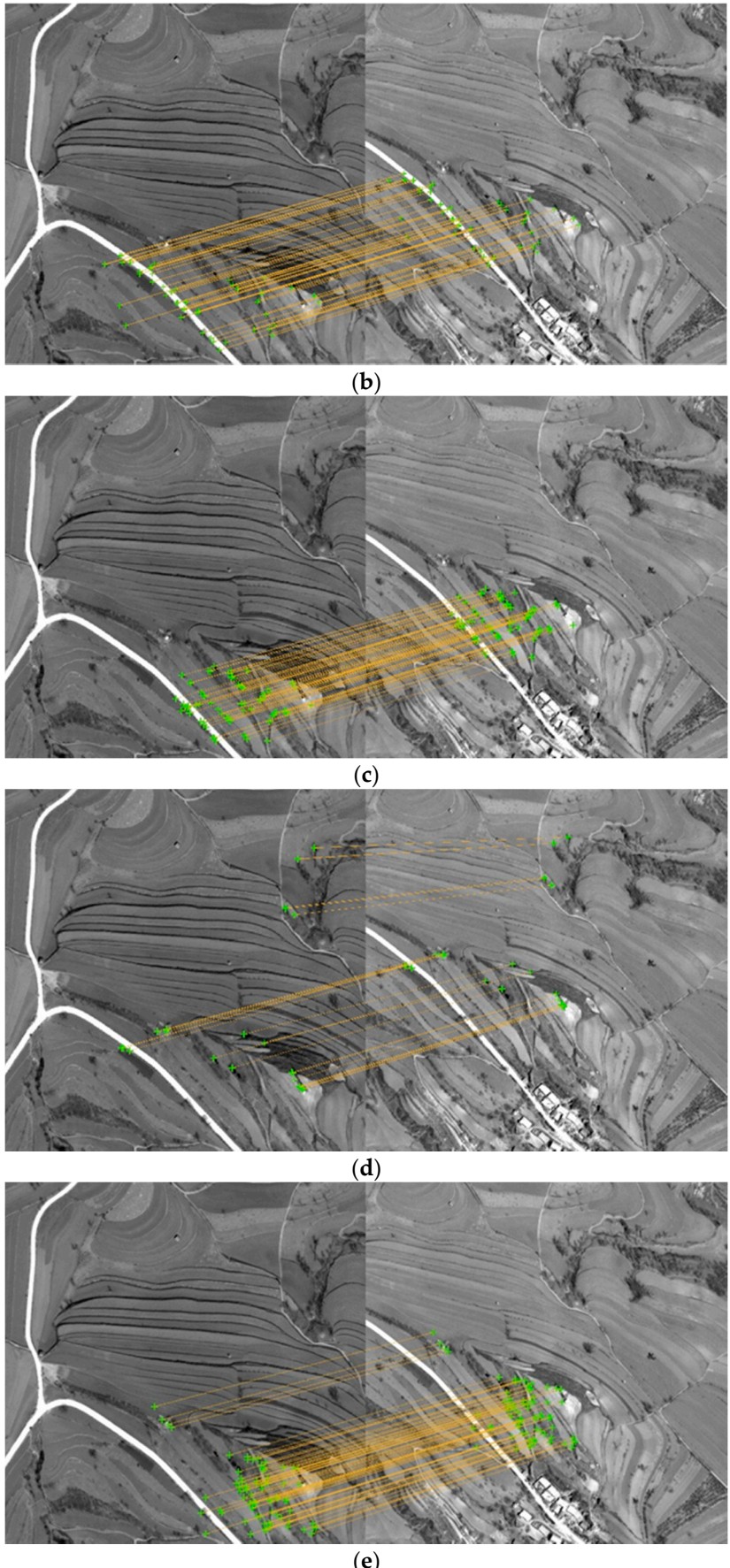

(**b**)

(**c**)

(**d**)

(**e**)

**Figure 6.** *Cont.*

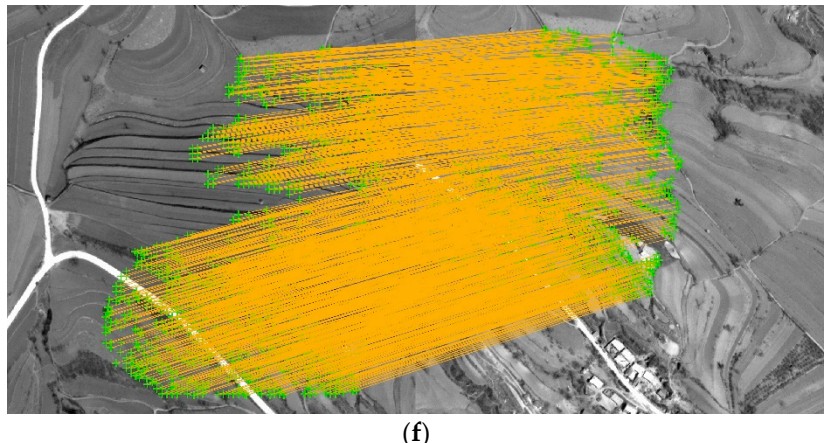

(**f**)

**Figure 6.** Comparing results between our method and other classic feature matching methods for the image pair 2: (**a**) SURF; (**b**) AKAZE; (**c**) ORB; (**d**) BRISK; (**e**) FAST; (**f**) our method.

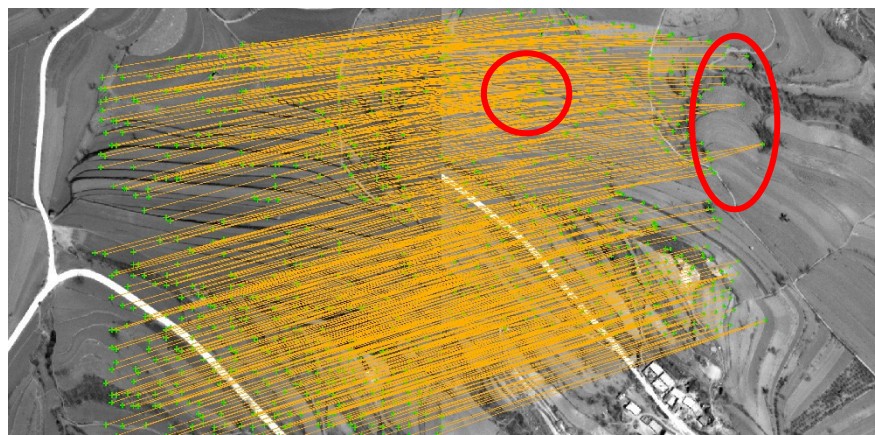

**Figure 7.** Matching results of our method without hyper-graph matching algorithm.

## 4. Conclusions

In this paper, a reweighted random walk based hyper-graph matching method is presented for the registration of high-resolution optical remote-sensing imagery. For the feature-based methods, they mainly utilize the radiometric information and the number of the correct matches is few and their distribution is uneven. This study proposes an efficient way to use the high-order structure information and radiometric information simultaneously and extends the hyper-graph matching method to the quasi-dense image matching domain. The proposed algorithm involves three steps: initial matching by a FBM method, two stage point matching, and outlier elimination. The initial matching is only carried out in the pyramid image of highest level to obtain the rough geometric relationship between the matching images. In the process of two-stage point matching, a rotation and scale invariant ABM is used to find candidate points for each feature point, and then by considering the candidate relationship between the matching points, a sparse high-order similarity tensor is efficiently built for hyper-graph matching, which helps to find correspondences between two sets of features. A local quadratic polynomial constraint framework is used to eliminate outliers of matched points. The acquired corresponding points are used as control points to carry out the image registration.

The proposed method has been evaluated using six datasets, which are captured by the high-resolution satellite optical sensors and cover different land types, such as mountain area, urban, suburb, and flat land. Furthermore, two datasets are covered by the cloud. These illustrated that the proposed method is highly adaptable to various situations and could be used for the practical high-resolution remote sensing image registration.

To demonstrate the advantage of the hyper-graph matching strategies, we have performed the experiment on image pair two with our method but without hyper-graph matching algorithms. Although the other strategies, such as rotation and scale invariant ABM, blunder rejection, and coarse-to-fine matching, enabled us to obtain large number of correct matches. However, when the image has repetitive textures, the intensity information is insufficient to find the correct matches. The hyper-graph matching algorithm simultaneously utilizes the radiometric and high-order geometric information to search the corresponding points, which contributes a lot to the improvement of matching robustness and success rate.

The relationship of the feature points are used, and a sparse high-order similarity tensor without losing any useful structure information is built, which enables us to overcome the computational burden and computer memory cost and introduce the hyper-graph matching algorithm to the remote sensing image registration.

To demonstrate the advantage of the proposed method, we compared it with the conventional matching algorithms, such as SURF, AKAZE, ORB, BRISK, and FAST. Four pairs of sub-images measuring $800 \times 800$ pixels are grabbed from the datasets of 1–4 and used for evaluation. Two criteria of recall and precision are used to evaluate the methods. From the experimental results, we find that the conventional methods can only derive very few feature points, which can only be used to derive global approximate geometric transformation. However, for our method, thousands of feature points can be derived, which guarantees the accurate image registration even for the difficult situations. The matching results can meet the image registration requirements of sub-pixel level accuracy.

However, as the high-order affinity tensor is needed to compute in our method, the computation time is longer than that of the compared methods, and how to improve the computational efficacy of the algorithm is our future work.

**Author Contributions:** L.D. reviewed and edited the original draft. Y.W. conceptualized the whole structure of the idea, developed the algorithm, and crafted the manuscript. Y.M. supervised the experiment and conducted the primary data analysis. H.L. and H.T. performed the experimental results analysis and revised the manuscript. All authors read and approved the final manuscript.

**Funding:** This work was supported by the National Natural Science Foundation of China (61301278), and by Natural Science Foundation of Hubei Province (2018CFB540), and by Philosophy and Social Science Foundation of Hubei Province (19Q062), and by Open Foundation of Hubei Collaborative Innovation Centre for High-efficient Utilization of Solar Energy (HBSKFM2014001), and China Scholarship Council (no. 201808420417) for Yingdan WU to conduct the research on which this paper is based at the Center for Spatial Information Science and Systems, George Mason University.

**Acknowledgments:** The research is supported by the Center for Spatial Information Science and Systems of George Mason University and thanks to C.M in Hubei provincial institute of land surveying and mapping for providing the experimental remote sensing imagery with cloud cover.

**Conflicts of Interest:** The authors declare no conflict of interest.

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
