# Peer review of "High-Resolution Optical Remote Sensing Image Registration via Reweighted Random Walk Based Hyper-Graph Matching"

_remotesensing, doi:10.3390/rs11232841_

Round 1

Reviewer 1 Report

This paper proposes a novel image registration method using a hyper-graph matching technique to extract a high number of accurate image point pair correspondences and then perform registration. They utilize UR-SIFT to perform initial matching at the highest pyramid level to extract approximate geometric relationships; then perform a two-stage point matching to find the possible matches and then a process to remove outliers. This process is iterated until the outliers are eliminated. The results show that the proposed method gets the best accuracy as compared to other image registration methods. 

The paper is detailed and well-written. I have few minor comments:

line 232 - a transpose seems missing a comparison with respect to computation time would be good to provide the recall of FAST seems consistently higher than the proposed method. What is the reason for that?

Author Response

Dear reviewer,

My response is in attachment.Thank you for your precious time.

Hope you everything goes well with you.

Best regards

Reviewer 2 Report

The paper describes a technique for image registration based on hyper graph matching using high-resolution satellite images. The authors applied the technique to 4 pairs of images, considering different types of coverage. The image registration was performed at subpixel level. In addition, the method was compared with 5 other current matching algorithms. In comparison, the results yielded a larger number of correct matches with more precision. Thus, the authors demonstrated that the technique contributes to improve the accuracy of image registration procedures. However, some points of the text still need to be improved before reaching publication. Comments are given below:

Comments

-Check the text for the term "radiation". Maybe "digital number" or "radiometric" is more appropriate.

- Pg. 3, section 2.1.2: It is not clear why you use the Forstner operator and then the SIFT since they generate different point sets. Why couldn't you just use SIFT? Was affine transformation calculated by least-squares? If so, how did you eliminate the outliers? Otherwise, outliers will affect estimates of transformation parameters. Please review this section.

- pg. 4, line 159: the authors could insert in the text the meanings of the hij parameters, that is, explain the effect of each parameter on the transformation.

- pg. 5, lines 167-168: ABM is applied, but what is the approximate size of the areas?

- pg. 7: Check if Equations 4 and 8 are typed correctly, or if there was a problem in converting to pdf file.

- pg. 9, line 309: Replace the term "rectify" with "resample" because you are not correcting the image.

- pg. 13, section 3.3: A more extensive discussion is missing in the results. For example, the authors could discuss the accuracy of their method in relation to the tested image content types (repetitive texture, relief, ...) by comparing them. Although the proposed method has not achieved the best results in the evaluation of the recall, a consideration of the results is also missing. Recall values were presented, but no discussion was presented.

- In pg 3, the authors comment that the proposed method has less computational cost, however, no performance evaluation was presented for comparing it with the other tested methods. In this case, the computational cost must be measured.

- The proposed method uses “Reweighted Random Walks”, but the authors did not explain how the weight matrix was determined in this application.

- An English revision is recommended to correct some mistakes.

Author Response

Dear reviewer,

My response is in attachment.Thank you for your precious time.

Hope you everything goes well with you.

Best regards.

Reviewer 3 Report

Please see an attached file for the review.

Paper # remotesensing-634696

Title: High-resolution Optical Remote Sensing Image Registration via Reweighted Random Walk based Hyper-graph Matching

Overview

In this study, the authors proposed a new framework of image registration for high-resolution remote sensing images. The image registration has been an important and essential issue for comparing satellite images obtained at different periods and obtained by (sometimes) different satellites. For such image registration, accurate feature matching and the number of matched pairs are crucial issues.

If my understanding is correct, the core contribution of this paper is proposing a new framework of feature matching by combination of a coarse-to-fine approach, a rotational and scale invariant ABM, and a hyper-graph matching technique, then the authors can achieve more accurate and robust feature matching. The resulted number of matched pairs was much higher than that from conventional methods (SURF, BRISK, etc..), as the authors argued.

Because the authors also achieved sub-pixel accuracy of the image registration for the high resolution images and the number of matched pairs from the proposed method shown in Figure 6 is surprisingly much better than the conventional methods, I think this study is potentially innovative worth to be published on Remote Sensing. However, unfortunately, I feel the authors did not conduct enough experiments for the publication in terms of variety of  weather conditions in remote sensing images, for instance, some parts in a remote sensing image are covered by clouds, or fog covers a whole image frame, resulting sever non-linear brightness change in a scene, while the authors seemed to test their method with only clear sky scenes.

The clouds and fog sometimes drastically worsen the performance of the image registration. Since it is possible that we have to use even such cloudy images when we consider real application of remote sensing images (e.g. assessing hazard damages immediately), it is worth to be investigated whether the proposed method works well even under such sever conditions.

Overall, although I feel that this study is innovative to be published in Remote Sensing due to  the good performance of the proposed method to the image registration of high-resolution remote sensing images, I strongly recommend the authors to add several experiments of image registration with cloudy and foggy (or thin clouds) scenes to proof the usefulness of their method in a real application as a major revision.

Main points:

- As I wrote in the overview of my review above, I strongly recommend the authors to add several experiments, which consider the existence of clouds and fog (or thin clouds), with the proposed method. The authors already tested their method with different brightness scenes, but all of them seemed clear sky scenes.

The reason why I have this concern is because most image-registration techniques recently proposed can treat such cloudy and foggy scenes by using an Earth-scale geometry constrain (e.g., Kouyama et al., 2017) or deep learning techniques (e.g., He et al., 2018; Dong et al., 2019, Figure 5c and 5h in Dong et al., 2019 may be a good example). Especially, deep learning techniques are very robust to non-linear intensity changes. Therefore, the readers may be interested in how much the proposed method is robust to such difficult scenes comparing to recent techniques (I think the authors know, SIFT, SURF, BRISK, AKAZE are no longer state-of-the-art techniques, they are rather conventional techniques). I understand the proposed method should be robust to non-linear brightness change due to utilizing both brightness and structure similarities.

Also, because clouds may affect to distribution of feature points a lot, this may change the performance of the proposed method which assumes well-distributed feature points. So the additional test with the consideration of clouds would be worth to try, I think.

References;

- Kouyama et al., Satellite Attitude Determination and Map Projection Based on Robust Image Matching, Remote Sens. 2017, 9, 90.

- Dong et al., Local Deep Descriptor for Remote Sensing Image Feature Matching. Remote Sens. 2019, 11, 430.

- He et al., Matching of Remote Sensing Images with Complex Background Variations via Siamese Convolutional Neural Network. Remote Sens. 2018, 10, 355.

Minor points:

- Abstract, Line 16

“an efficient hyper-graph matching method is proposed, …”

Because in the proposed method, the hyper-graph matching is a part of the framework, it should be:

an efficient method utilizing hyper-graph matching algorithm is proposed, …

- Abstract Line 30

“The experiments show that the proposed method outperforms state-of-the-art matching algorithms such as SURF, AKAZE, ORB, BRISK and FAST”

As I wrote in the main point, SURF, AKAZE, ORB, BRISK and FAST are not state-of-the-art methods, but rather conventional or traditional methods. We may say them as “the methods commonly/widely used” or similar expression. I found the authors said “Compared with traditional feature matching algorithms, including SURF, AKAZE, ORB, BRISK and FAST,…” in the conclusion section. Please check the expression.

- Lines 68-69

Kouyama et al (2017) and Sugimoto et al (2018) utilized observation geometry as a constrain in their RANSAC steps and achieved robust performance of outlier rejection even under the cloudy condition. Please add the references for mentioning there have been some trials for improving the RANSAC approach in the remote sensing context.

- Line 79

It should be better to refer recent studies which utilize deep learning techniques for image registration of remote sensing images as another trend of the SIFT-like approach. For instance, Dong et al and He et al.

- Lines 130-137

Because there is no information about “Pyramid level” in this overview section (2.1.2) although figure 1 includes the loop for Pyramid levels, please mention here about “pyramid levels” and pyramid strategy. It may be better that descriptions in Line 141-146 is moved to this overview section.

- Lines 163-164

What is a typical size of the virtual grid cells and how many virtual grid cells you set? Are they aligned regularly? It would be good information when readers follow your idea.

- Section 2.3.1

I am interested in the performance of the rotation and scale invariant ABM. I feel it might be possible the authors will get enough number of feature points from only the step of the rotation and scale invariant ABM and an outlier rejection procedure, instead of applying the two-step matching. Have the authors tried image registration without the hyper-graph matching step? If the rotation and scale invariant ABM is a key contributor to the robustness of the proposed method, it should be good information for the readers.

- Figure 4

It is hard to confirm the proposed method actually achieved the sub-pixel level image registration or not from panels in Figure 4. Please use additional zoom panels for each example to show the detail of the image registration. Maybe Figure 4 in Dong et al is a good example.

- Table 3

I feel the scores (especially precision) from the conventional methods in this table were too low, because the tested image size (800x800 pixels) is enough large and they seem to have a lot of textures. Of course, periodical patterns and feature distortion that occur frequently in high-resolution images may decrease the performance of the conventional methods. But to confirm the fairness of the comparison, please provide detail information how the authors applied the conventional methods to the 800x800 pixel images.

In addition, here the author could add a result from the rotation and scale invariant ABM + outlier rejection.

Typos:

- Line 267, and Line 451

IBM” should be ABM

Author Response

Dear reivewer,

My response is in attachment.Thank you for your precious time.

Hope you everything goes well with you.

Best regards

Round 2

Reviewer 2 Report

The authors performed the corrections. The questions were also answered and clarifications were inserted in the text. Therefore, in my opinion, the paper has been improved and can be accepted.

Author Response

Dear reviewer,

Thank you for your precious suggestion.

My manuscript has been well improved.

Hope you everything goes well with you.

Best regards

Reviewer 3 Report

I feel the revised version is well improved from the original version, and I satisfied the authors’ responses for my comments and suggestions. However, I feel small corrections in the revised parts are still needed before the publication.

Minor comments are below:

Lines 86-87

However, all these feature-based matching algorithms only depending on the intensity similarity of feature points, they are prone to produce incorrect matches.

Now the authors referred some works based on deep learning methods [33, 34], and because the deep learning with convolutional neural networks can consider spatial structures around a target point, I feel above description needs a small correction. Please revise this description. It could be:

“However, all these feature-based matching algorithms only depending on the intensity similarity of feature points, or the spatial similarity of only nearby features, they are prone to produce incorrect matches.”

Lines 135-136:

Before starting matching, the pyramid images are generated.

Since the word “the pyramid image” without any context provides the image of just “pyramid” in the literature. It could be:

“Before starting matching, the image pyramid with three levels is generated from original image in which different levels have different spatial resolutions (see Section 2.2).”

Discussion section:

In general, Discussion section is used for discussing some questions that rise through experiments, or used for providing additional information to investigate why the proposed method works well than previous methods quantitatively, or so (for instance, one possible discussion topic is how “triple similarity tensor” contributes the accurate matching with some experiments with and without the triple similarity).

Although the authors added the Discussion section, I feel it is hard to see the sentences in this section as discussion, unfortunately, rather these sentences seem conclusion remarks.

So I would recommend the authors to just merge the sentences that the authors newly added in the discussion section with the sentences in the conclusion section, or to add some quantitative evaluations for the proposed method if the authors want to add the discussion section.

Author Response

Dear reviewer,

My response is in attachment.Thank you for your precious time.

My manuscript has been well improved on account of your valuable suggestion.

Hope you everything goes well with you.

Best regards
